# Human Holliday junction resolvase GEN1 uses a chromodomain for efficient DNA recognition and cleavage

**Shun-Hsiao Lee[1], Lissa Nicola Princz[2], Maren Felizitas Klügel[1], Bianca Habermann[3], Boris Pfander[2], Christian Biertümpfel[1]***

[1]Department of Structural Cell Biology, Molecular Mechanisms of DNA Repair, Max Planck Institute of Biochemistry, Martinsried, Germany; [2]Department of Molecular Cell Biology, DNA Replication and Genome Integrity, Max Planck Institute of Biochemistry, Martinsried, Germany; [3]Computational Biology, Max Planck Institute of Biochemistry, Martinsried, Germany

**Abstract** Holliday junctions (HJs) are key DNA intermediates in homologous recombination. They link homologous DNA strands and have to be faithfully removed for proper DNA segregation and genome integrity. Here, we present the crystal structure of human HJ resolvase GEN1 complexed with DNA at 3.0 Å resolution. The GEN1 core is similar to other Rad2/XPG nucleases. However, unlike other members of the superfamily, GEN1 contains a chromodomain as an additional DNA interaction site. Chromodomains are known for their chromatin-targeting function in chromatin remodelers and histone(de)acetylases but they have not previously been found in nucleases. The GEN1 chromodomain directly contacts DNA and its truncation severely hampers GEN1's catalytic activity. Structure-guided mutations in vitro and in vivo in yeast validated our mechanistic findings. Our study provides the missing structure in the Rad2/XPG family and insights how a well-conserved nuclease core acquires versatility in recognizing diverse substrates for DNA repair and maintenance.

***For correspondence:**
biertuempfel@biochem.mpg.de

**Competing interests:** The authors declare that no competing interests exist.

## Introduction

Homologous recombination (HR) is a fundamental pathway ensuring genome integrity and genetic variability (*Heyer, 2015*). In mitotic cells, double-strand breaks (DSBs) can be repaired by HR using the sister chromatid as a template to restore the information in the complementary double strand. In meiosis, the repair of programmed DSBs by HR and the formation of crossovers are crucial to provide physical linkages between homologs and to segregate homologous chromosomes. Furthermore, HR during meiosis creates sequence diversity in the offspring through the exchange between homologs (*Petronczki et al., 2003*; *Sarbajna and West, 2014*).

HR proceeds by pathways that may lead to the formation of DNA four-way junctions or Holliday junctions (HJs) that physically link two homologous DNA duplexes (*Heyer, 2015*; *Holliday, 1964*; *Schwacha and Kleckner, 1995*; *Szostak et al., 1983*). Faithful removal of HJs is critical to avoid chromosome aberrations (*Wechsler et al., 2011*) and cells have evolved sophisticated measures to disentangle joint molecules. One basic mechanism is resolution mediated by HJ resolvases that introduce precise symmetrical nicks into the DNA at the branch point. Nicked DNA strands are then rejoined by endogenous ligases leading to fully restored or recombined DNA strands. This mechanism is well studied for bacterial and bacteriophage resolvases such as *Escherichia coli* RuvC, T7 endonuclease I, T4 endonuclease VII (*Benson and West, 1994*; *Lilley and White, 2001*). These resolvases operate as dimers and show a large degree of conformational flexibility in substrate

**eLife digest** Factors like ultraviolet radiation and harmful chemicals can damage DNA inside living cells, which can lead to breaks that form across both strands in the DNA double helix. "Homologous recombination" is one of the major mechanisms by which cells repair these double-strand breaks. During this process, the broken DNA interacts with another undamaged copy of the DNA to form a special four-way structure called a "Holliday junction". The intact DNA strands are then used as templates to repair the broken strands. However, once this has occurred the Holliday junction needs to be 'resolved' so that the DNA strands can disentangle.

One way in which Holliday junctions are resolved is through the introduction of precise symmetrical cuts in the DNA at the junction by an enzyme that acts like a pair of molecular scissors. Re-joining these cut strands then fully restores the DNA. Enzymes that generate the cuts in DNA are called nucleases, and the nuclease GEN1 is crucial for resolving Holliday junctions in organisms such as fungi, plants and animals. GEN1 belongs to a family of enzymes that act on various types of DNA structures that are formed either during damage repair, DNA duplication or cell division. However, GEN1 is the only enzyme in the family that can also recognize a Holliday junction and it was unclear why this might be.

Lee et al. have now used a technique called X-ray crystallography to solve the three-dimensional structure of the human version of GEN1 bound to a Holliday junction. This analysis revealed that many features in GEN1 resemble those found in other members of the same nuclease family. These features include two surfaces of the protein that bind to DNA and are separated by a wedge, which introduces a sharp bend in the DNA. However, Lee et al. also found that GEN1 contains an additional region known as a "chromodomain" that further anchors the enzyme to the DNA. The chromodomain allows GEN1 to correctly position itself against DNA molecules, and without the chromodomain, GEN1's ability to cut DNA in a test tube was severely impaired. Further experiments showed that the chromodomain was also important for GEN1's activity in yeast cells growing under stressed conditions.

The discovery of a chromodomain in this human nuclease may provide many new insights into how GEN1 is regulated, and further work could investigate if this chromodomain is also involved in binding to other proteins.

recognition and in aligning both active sites for coordinated cleavage. Interestingly, T4 endonuclease VII and RuvC reach into and widen the DNA junction point whereas T7 endonuclease I binds DNA by embracing HJs at the branch point (*Biertümpfel et al., 2007*; *Górecka et al., 2013*; *Hadden et al., 2007*).

In eukaryotes, HR is more complex and tightly regulated. In somatic cells, HJ dissolution by a combined action of a helicase and a topoisomerase (BLM-TOPIIIα-RMI1-RMI2 complex in humans) is generally the favored pathway, possibly to restore the original (non-crossover) DNA arrangement (*Cejka et al., 2010*, *2012*; *Ira et al., 2003*; *Putnam et al., 2009*; *Wu and Hickson, 2003*). In contrast, HJ resolution generates crossover and non-crossover arrangements depending on cleavage direction. Several endonucleases such as GEN1, MUS81-EME1, and SLX1-SLX4 have been implicated as HJ resolvases in eukaryotes (*Andersen et al., 2011*; *Castor et al., 2013*; *Fekairi et al., 2009*; *Garner et al., 2013*; *Ip et al., 2008*; *Muñoz et al., 2009*; *Svendsen and Harper, 2010*; *Svendsen et al., 2009*; *Wyatt et al., 2013*). Interestingly, these resolvases are not structurally related and have different domain architectures, giving rise to variable DNA recognition and regulation mechanisms. The interplay between resolution and dissolution mechanisms is not fully understood yet, however, cell cycle regulation of resolvases seems to play an important role (*Blanco et al., 2014*; *Chan and West, 2014*; *Eissler et al., 2014*; *Matos et al., 2011*).

GEN1 belongs to the Rad2/XPG family of structure-selective nucleases that are conserved from yeast to humans (*Ip et al., 2008*; *Lieber, 1997*; *Yang, 2011*). The Rad2/XPG family has four members with different substrate preferences that function in DNA maintenance (*Nishino et al., 2006*; *Tsutakawa et al., 2014*). They share a conserved N-terminal domain (XPG-N), an internal domain (XPG-I) and a 5'->3' exonuclease C-terminal domain containing a conserved helix-hairpin-helix motif.

C-terminal to the nuclease core is a regulatory region that is diverse in sequence and predicted to be largely unstructured. Although the catalytic cores are well conserved in the superfamily, substrate recognition is highly diverse: XPG/Rad2/ERCC5 recognizes bubble/loop structures during nucleotide-excision repair (NER), FEN1 cleaves flap substrates during Okazaki fragment processing in DNA replication, EXO1 is a 5'->3' exonuclease that is involved in HR and DNA mismatch repair (MMR) and GEN1 recognizes Holliday junctions (*Grasby et al., 2012*; *Ip et al., 2008*; *Nishino et al., 2006*; *Tomlinson et al., 2010*; *Tsutakawa et al., 2014*). A common feature of the superfamily is their inherent ability to recognize flexible or bendable regions in the normally rather stiff DNA double helix. Interestingly, GEN1 shows versatile substrate recognition accommodating 5' flaps, gaps, replication fork intermediates and Holliday junctions (*Ip et al., 2008*; *Ishikawa et al., 2004*; *Kanai et al., 2007*). According to the current model, however, the primary function of GEN1 is HJ resolution (*Garner et al., 2013*; *Sarbajna and West, 2014*; *West et al., 2015*) and it is suggested to be a last resort for the removal of joint molecules before cytokinesis (*Matos et al., 2011*).

To date, structural information is available for all members of the family but GEN1 (*Miętus et al., 2014*; *Orans et al., 2011*; *Tsutakawa et al., 2011*). A unified feature of these structures is the presence of two DNA-binding interfaces separated by a hydrophobic wedge. This wedge is composed of two protruding helices that induce a sharp bend into flexible DNA substrates. Rad2/XPG family members also share a helix-two-turn-helix (H2TH) motif that binds and stabilizes the uncleaved DNA strand downstream of the catalytic center. However, the comparison of DNA recognition features within the Rad2/XPG family has been hampered because of the lack of structural information on GEN1.

To understand the molecular basis of GEN1's substrate recognition, we determined the crystal structure of human GEN1 in complex with HJ DNA. In combination with mutational and functional analysis using in vitro DNA cleavage assays and in vivo survival assays with mutant yeast strains, we highlight GEN1's sophisticated DNA recognition mechanism. We found that GEN1 does not only have the classical DNA recognition features of Rad2/XPG nucleases, but also contains an additional DNA interaction site mediated by a chromodomain. In the absence of the chromodomain, GEN1's catalytic activity was severely impaired. This is the first example showing the direct involvement of a chromodomain in a nuclease. Our structural analysis gives implications for a safety mechanism using an adjustable hatch for substrate discrimination and to ensure coordinated and precise cleavage of Holliday junctions.

## Results

### Structure determination and architecture of the GEN1-DNA complex

In order to structurally characterize human GEN1, we crystallized the catalytically inactive variant GEN1$^{2\text{-}505\ D30N}$, denoted GEN1 for simplicity, in complex with an immobile Holliday junction having arm lengths of 10 bp (*Figure 1*). The structure was determined experimentally and refined up to 3.0 Å resolution with an $R_{free}$ of 0.25 (*Table 1*). The HJ crystallized bridging between two protein monomers in the asymmetric unit (*Figure 1—figure supplement 1*). The overall structure of GEN1 resembles the shape of a downwards-pointing right hand with a 'thumb' extending out from the 'palm' and the DNA is packed against the ball of the thumb (*Figure 1*). The palm contains the catalytic core, which is formed by intertwined XPG-N and XPG-I domains (*Figure 1A/B*, green). They consist of a seven-stranded β-sheet in the center surrounded by nine helices harboring the conserved active site (*Figure 1B/D*, orange). The catalytic residues form a cluster of negatively charged residues (D30, E75, E134, E136, D155, D157, D208) that were originally identified by mutational analysis (*Ip et al., 2008*; *Lee et al., 2002*; *Wakasugi et al., 1997*) and are conserved in other Rad2/XPG family members (*Figure 1B/C* and *Figure 2*). The XPG-I domain is followed by a 5'->3' exonuclease C-terminal domain (EXO; *Figure 1B/D*, blue). The EXO domain consists of a helix-two-turn-helix (H2TH) motif (helices α10-α11) accompanied by several α-hairpins (α12-α13 and α14-α15). A similar arrangement is also found in other proteins, which use a H2TH motif for non-sequence specific DNA recognition (*Tomlinson et al., 2010*). The EXO domain in GEN1 has a 78 amino acid insertion (residues 245–322), of which only helix α12b (residues 308–322) is ordered in the structure (*Figure 1A*, gray and *Figure 2*). Helix α12b packs loosely with the H2TH helices (α10-α11) and helix α12 at the 'finger' part of GEN1. Yeast Rad2, a homolog of human XPG, also contains helix α12b,

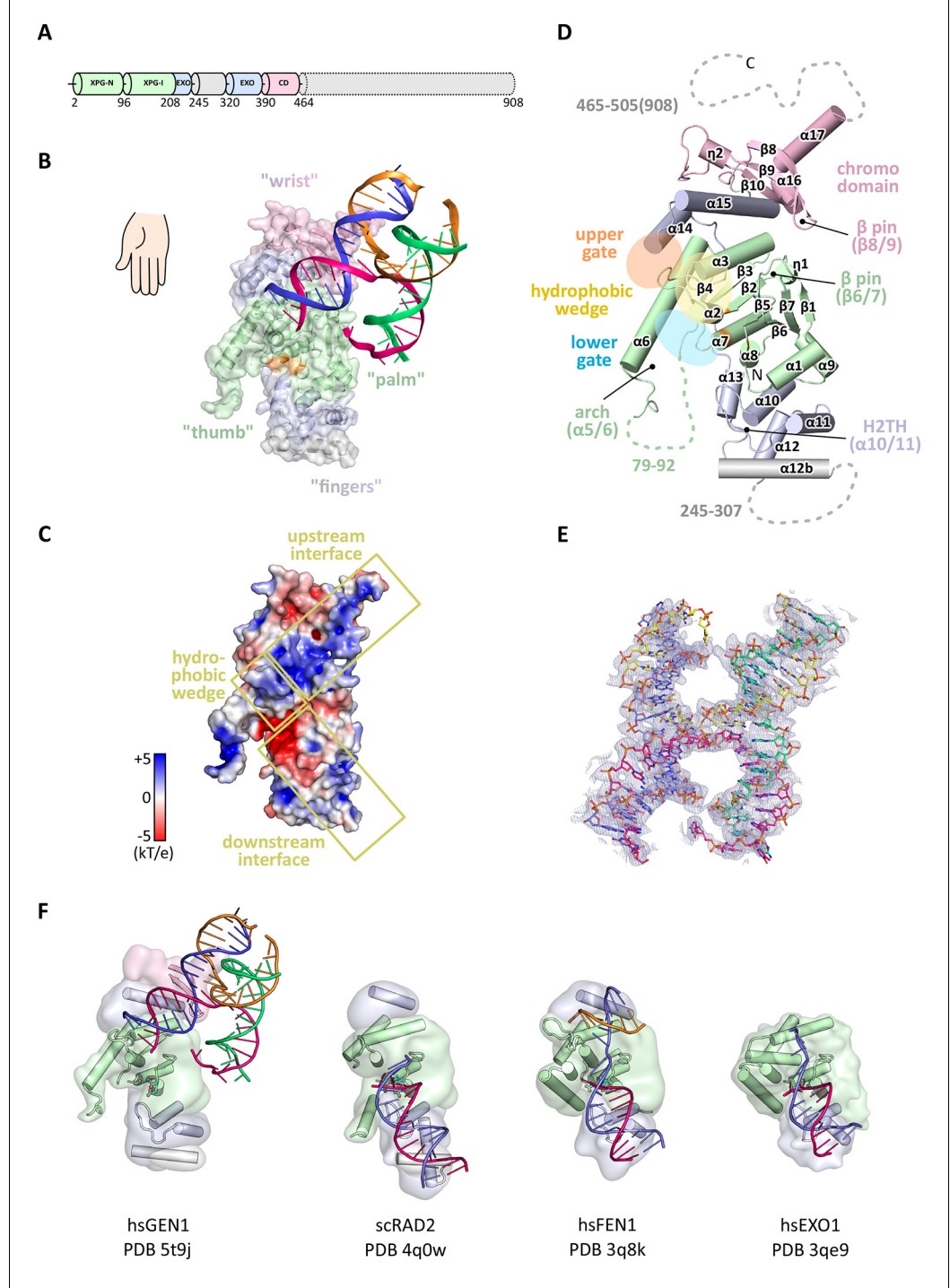

**Figure 1.** Architecture of human GEN1. (**A**) Domain architecture of human GEN1. The structurally unknown regulatory domain (residues 465–908) is shown with dotted lines. (**B**) Overview of the catalytic core of GEN1 in complex with HJ DNA. The protein resembles the shape of a downwards-pointing right hand with helix α6 as the thumb. The protein is depicted in half transparent surface representation with secondary structure elements underneath. The DNA is shown in ladder representation with individual strands in different colors. The coloring of GEN1 follows domain boundaries: intertwining XPG-N and XPG-I in green, 5'->3' exonuclease C-terminal domain (EXO) in blue, chromodomain in pink, unassigned regions in gray. Active site residues (E134, E136, D155, D157) are highlighted in orange. (**C**) Electrostatic surface potential of GEN1. The coloring follows the potential from -5 (red) to +5 kT/e (blue). The DNA-binding interfaces and the position of the hydrophobic wedge are marked in yellow. (**D**) Secondary structure elements of the catalytic core of GEN1 in cartoon representation with the same

*Figure 1 continued on next page*

*Figure 1 continued*

colors as before. Dotted lines represent parts that are not resolved in the crystal structure. The numbering follows a unified scheme for the Rad2/XPG family (compare *Figure 2*) for α-helices, β-sheets and $3_{10}$-helices (η). (E) Experimental electron density map (autoSHARP, solvent flattened, contoured at 1σ) drawn around the HJ in the GEN1 complex. The DNA model is shown in ball-stick representation with carbon atoms of individual strands in different colors (yellow, light blue, magenta, green) and oxygen atoms in red, phosphor atoms in orange, nitrogen atoms in dark blue. (F) Structural comparison of Rad2/XPG family nucleases. Proteins are shown in a simplified surface representation with important structural elements in cartoon representation and DNA in ladder representation. The color scheme is the same as in B. *Figure 1—figure supplement 1* shows the content of the asymmetric unit.

The following figure supplement is available for figure 1:

**Figure supplement 1.** Content of the asymmetric unit of the GEN1-HJ crystal.

---

and it shows a similar arrangement as in GEN1 (*Figure 1F*). The EXO domain sandwiches the XPG-N/I domains with a long linker reaching from the bottom 'fingers' (α10-α13) along the backside of GEN1 to the top of the XPG-N/I domains at the 'wrist' (α14-α15). A structure-based sequence alignment of the nuclease core of human GEN1, FEN1, EXO1 and yeast Rad2 proteins with functional annotations relates sequence conservation to features in the Rad2/XPG family (*Figure 2*). The comparison with members in the Rad2/XPG identified two DNA binding interfaces and a hydrophobic wedge (ball of the thumb) that separates the upstream and the downstream interface (*Figure 1C/D* and compare *Figure 1F*). GEN1 has two prominent grooves close to the hydrophobic wedge, which we termed upper and lower gate or gateway for comparison (*Figure 1D*, orange and blue ellipses, respectively).

Notably, a small globular domain (residues 390–464) was found extending the GEN1 nuclease core at the wrist (*Figure 1*, pink). A DALI search (*Holm and Rosenström, 2010*) against the Protein Data Bank (PDB) identified this domain as a chromodomain (chromatin organization modifier domain). The domain has a chalice-shaped structure with three antiparallel β-strands packed against a C-terminal α-helix and it forms a characteristic aromatic cage. The opening of the chalice abuts helix α15 from the EXO domain.

## GEN1 has a conserved chromodomain with a closed aromatic cage

Chromodomains are found in many chromatin-associated proteins that bind modified histone tails for chromatin targeting (reviewed in *Blus et al., 2011*; *Eissenberg, 2012*; *Yap and Zhou, 2011*), but it has not previously been associated with nucleases. To understand the significance of the chromodomain for the function of GEN1, we first examined if the chromodomain is conserved in GEN1 homologs using HMM-HMM (Hidden Markov Models) comparisons in HHPRED (*Söding et al., 2005*). We found that the chromodomain in GEN1 is conserved from yeast (Yen1) to humans (*Figure 3A*). The only exception is *Caenorhabditis elegans* GEN1, which has a much smaller protein size of 443 amino acids compared to yeast Yen1 (759 aa) or human GEN1 (908 aa).

To further compare the structural arrangement of the aromatic cage in human GEN1 with other chromodomains, we analyzed the best matches from the DALI search (*Figure 3B*). We found many hits for different chromo- and chromo-shadow domains with root mean square deviations between 1.9 and 2.8 Å (compare *Figure 3—source data 1*). A superposition of the aromatic cage of the five structurally most similar proteins with GEN1 (*Figure 3C*) showed that residues W418, T438, and E440 are well conserved, whereas two residues at the rim of the canonical binding cleft are changed from phenylalanine/tyrosine to a leucine (L397) in one case and a proline (P421) in another (*Figure 3C*). Instead, Y424 occupies the space proximal to P421, which is about 1.5 Å outwards of the canonical cage and widens the GEN1 cage slightly. The substitution of phenylalanine/tyrosine to leucine is also found in CBX chromo-shadow domains (see below); however, the rest of the GEN1 aromatic cage resembles rather chromodomains.

Chromodomains often recognize modified lysines through their aromatic cage thus targeting proteins to chromatin (reviewed in *Blus et al., 2011*; *Eissenberg, 2012*; *Yap and Zhou, 2011*). Given the conserved aromatic cage in GEN1, we tested the binding to modified histone tail peptides

**Table 1.** Data collection and refinement statistics.

| Data Set | G505-4w006native | G505-4w006Ta peak | G505-4w006SeMet peak |
|---|---|---|---|
| **Diffraction Data Statistics** | | | |
| Synchrotron Beamline | SLS PXII | SLS PXII | SLS PXII |
| Wavelength | 0.99995 | 1.25473 | 0.97894 |
| Resolution (Å) | 75-3.0 | 75.4-3.8 | 43.6-4.4 |
| Space Group | P 3$_2$ | P 3$_2$ | P 3$_2$ |
| Cell dimensions | | | |
| a (Å) | 86.94 | 87.06 | 87.11 |
| b (Å) | 86.94 | 87.06 | 87.11 |
| c (Å) | 200.72 | 201.30 | 199.69 |
| α (°) | 90 | 90 | 90 |
| β (°) | 90 | 90 | 90 |
| γ (°) | 120 | 120 | 120 |
| $I/\sigma I$* | 18.4 (1.9) | 27.49 (5.83) | 16.58 (3.82) |
| Completeness (%)* | 99.8 (98.8) | 99.6 (97.3) | 97.3 (83.3) |
| Redundancy* | 6.3 | 10.2 | 5.1 |
| $Rsym$ (%)* | 6.2 (90.7) | 7.7 (42.2) | 6.9 (43.4) |
| **Refinement Statistics** | | | |
| Resolution (Å) | 75-3.0 | | |
| Number of Reflections | 33933 | | |
| $R_{work}/R_{free}$ | 0.199/0.241 | | |
| Number of Atoms | | | |
| Protein | 6298 | | |
| DNA | 1589 | | |
| Water/Solutes | 27 | | |
| B-factors | | | |
| Protein | 123.4 | | |
| DNA | 150.2 | | |
| Water/Solutes | 92.6 | | |
| R.M.S Deviations | | | |
| Bond lengths (Å) | 0.010 | | |
| Bond Angles (°) | 0.623 | | |
| Ramachandran Plot | | | |
| Preferred | 753 (97.9 %) | | |
| Allowed | 16 (2.1%) | | |

*Values for the highest resolution shell are shown in parenthesis

(*Figure 3C/D*). However, we did not detect any binding despite extensive efforts using various histone tail peptides in pull-down assays, microscale-thermophoresis (MST) or fluorescence anisotropy measurements (compare *Figure 3—source data 2* and *Figure 3—figure supplement 2*). Our structure shows that the aromatic cage is closed by helix α15 (*Figure 3E* blue/pink), which has a hydrophobic interface towards the aromatic cage with residues L376, T380, and M384 reaching into it (compare *Figure 4F*). This potentially hampers the binding of the tested peptides in this conformation under physiological conditions.

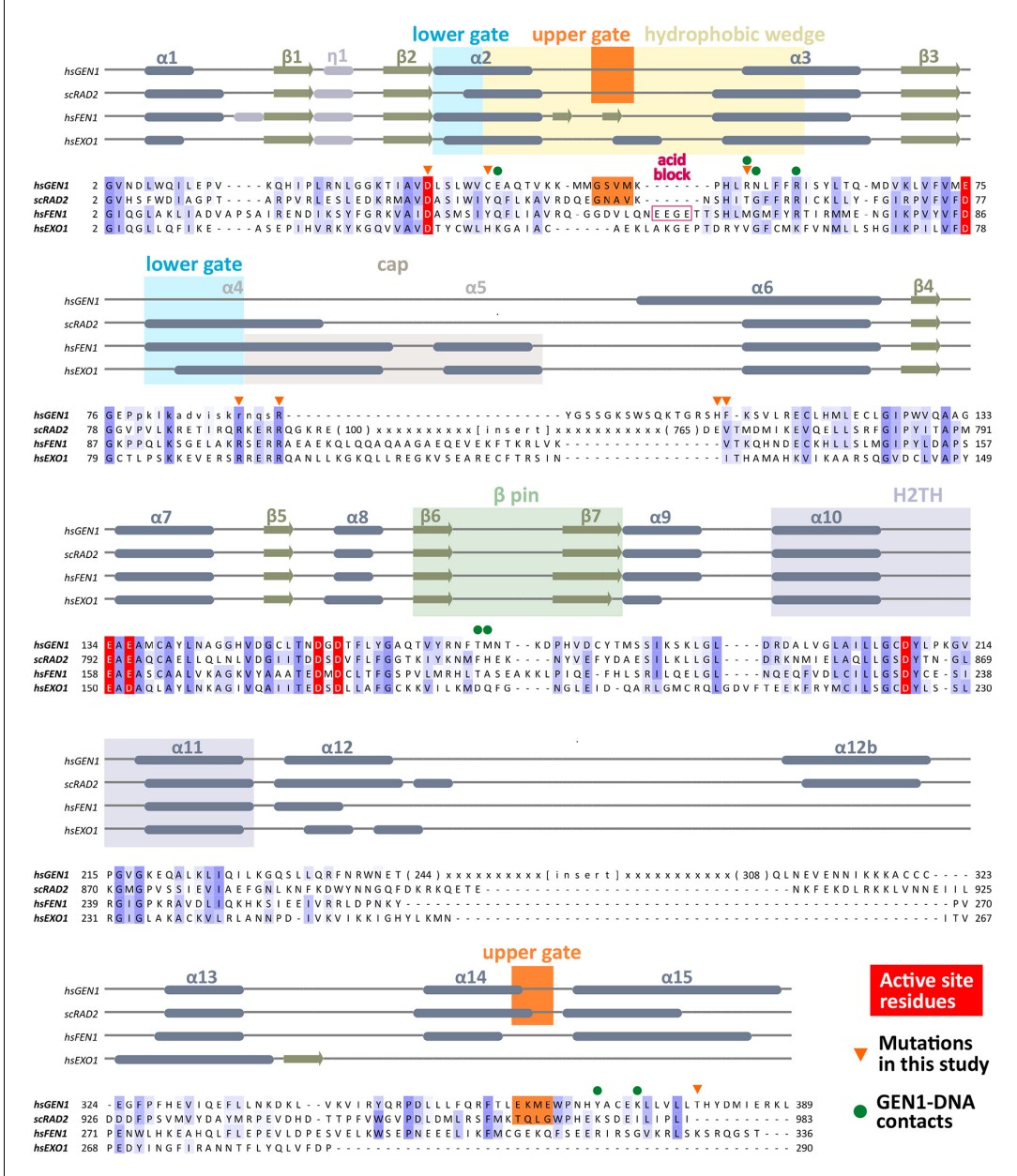

**Figure 2.** Alignment of the nuclease cores of Rad2/XPG-family proteins. The alignment is based on known crystal structures: human GEN1 (PDB 5t9j, this study), yeast Rad2 (PDB 4q0w), human FEN1 (PDB 3q8k), human EXO1 (3qe9). Secondary structure elements are depicted on top of the sequence with dark blue bars for α-helices, light blue bars for 3₁₀-helices and green arrows for β-sheets. The numbering follows a unified scheme for the superfamily. Functional elements are labeled and described in the main text. Sequences are colored by similarity (BLOSUM62 score) and active site residues are marked in red. Mutations analyzed in this study are marked with an orange triangle and DNA contacts found in the human GEN1–HJ structure have a dark green dot. Disordered or missing parts in the structures are labeled in small letters or with x.

## The GEN1 chromodomain is distantly related to CBX and CDY chromodomains

To explore the functional role of the GEN1 chromodomain, we evaluated its similarity to other chromodomains by comparing all of the 46 known human chromodomains from 34 different proteins. We made pairwise comparisons with HHPRED, PSIBLAST, combined the alignments and generated a phylogenetic tree (*Figure 3F* and *Figure 3—figure supplement 1*). The analysis showed a tree

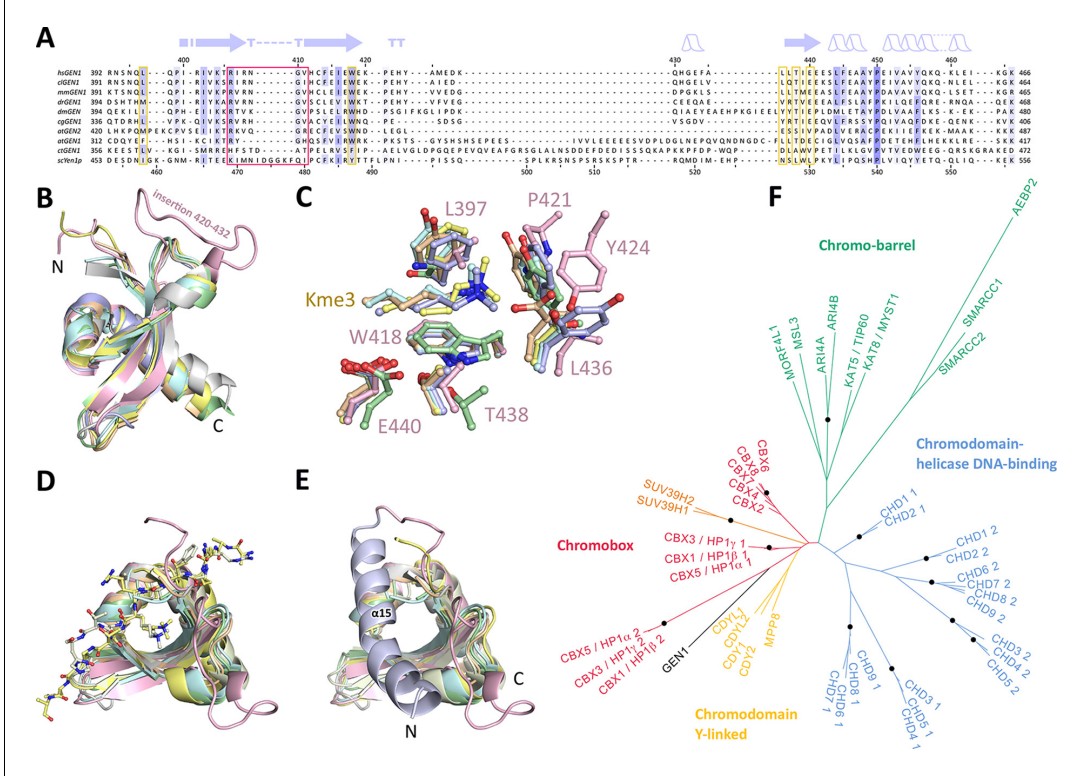

**Figure 3.** Chromodomain comparison. (**A**) Sequence alignment of GEN1 chromodomains from different organisms: hsGEN1 (*Homo sapiens*), clGEN1 (*Canis lupus*), mmGEN1 (*Mus musculus*), drGEN1 (*Danio rerio*), atGEN1/2 (*Arabidopsis thaliana*), cgGEN1 (*Crassostrea gigas*), scYEN1 (*Saccharomyces cerevisiae*). The presence of a chromodomain is conserved from yeast to human with *Caenorhabditis elegans* as an exception. Secondary structure elements of the GEN1 chromodomain are shown on top. The sequence coloring is based on a similarity matrix (BLOSUM62). The corresponding positions of the DNA-interaction site in human GEN1 is marked with a red box and residues of the aromatic cage are highlighted with a yellow box. (**B**) GEN1 has a canonical chromodomain fold of three antiparallel beta-sheets packed against an α-helix. (**C**) The arrangement of the aromatic cage in GEN1 is comparable to other chromodomains but less aromatic and slightly larger. (**D**) The superposition of different chromodomains places cognate binding peptides of hsMPP8 and mmCBX7 (and others) into the aromatic cage. (**E**) The aromatic cage of GEN1 is closed by helix α15. Panels **B–D** show the chromodomains of hsGEN1 (pink, PDB 5t9j), hsCBX3 (gray, PDB 3kup) hsSUV39H1 (green, PDB 3mts), hsMPP8 (yellow, PDB 3lwe), dmHP1a (orange, chromo shadow PDB 3p7j), dmRHINO (cyan, PDB 4quc/3r93), mmCBX7 (light blue, PDB 4x3s; compare *Figure 3—source data 1*). (**F**) Phylogenetic tree of all known human chromodomains. GEN1 is distantly related to the CBX chromo-shadow domains and CDY chromodomains. The corresponding alignment for calculating the phylogenetic tree is shown in *Figure 3—figure supplement 1*. GEN1 is colored in black, chromobox (CBX) proteins are colored in red, interspersed by SUV39H histone acetylases (orange) and chromodomain Y-linked (CDY) proteins (yellow). Chromo-barrel domain proteins are colored in green and chromodomain-helicase DNA-binding (CHD) proteins are in blue. Chromodomains and chromo-shadow domains from the same protein are labeled with 1 and 2, respectively. Stable branches with boostrap values equal or higher than 0.8 are marked with a black dot. The binding of the GEN1 chromodomain to a set of histone peptides was tested but no interaction was detected (*Figure 3—source data 2* and *Figure 3—figure supplement 2*).

The following source data and figure supplements are available for figure 3:

**Source data 1.** Proteins found in a DALI search.

**Source data 2.** N-terminally fluorescein-labeled peptides used for chromodomain binding assays.

**Figure supplement 1.** Sequence alignment of all known human chromodomains.

**Figure supplement 2.** Histone peptide pull-down assay.

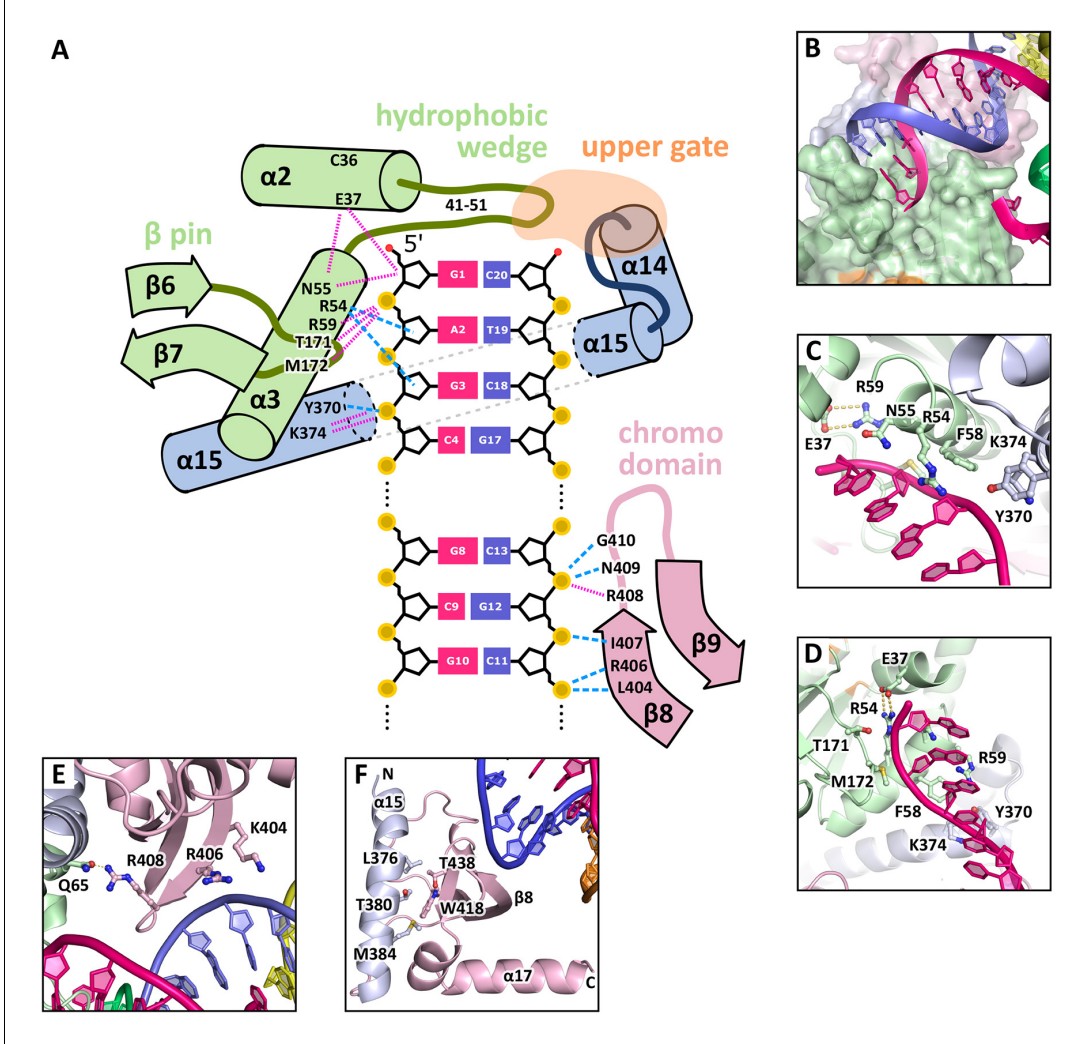

**Figure 4.** DNA interactions in the GEN1-DNA complex. (**A**) Schematic of the GEN1-DNA interactions at the upstream interface. The coloring is the same as in *Figure 1*. The nuclease core (green and blue) interacts with the uncleaved strand and the chromodomain (pink) contacts the complementary strand. Hydrogen bonds are shown with blue dashed lines and van-der-Waals contacts are in red dotted lines. (**B**) Interactions at the hydrophobic wedge. The end of the DNA double helix docks onto the hydrophobic wedge formed by helices α2 and α3. (**C/D**) Interactions with the uncleaved strand in two views. All key residues form sequence-independent contacts to the DNA backbone. R54 reaches into the minor groove of the DNA. The complementary DNA strand has been removed for clarity (**E/F**) Interactions of the chromodomain with the complementary strand in two views. The backbone of residues 406–410 (β-hairpin β8-β9) abuts the DNA backbone. R406 has a supporting role in the interaction and R408 forms a polar interaction with Q65, which establishes a connection between the chromodomain and the nuclease core. Helix α15 makes hydrophobic interactions with the aromatic cage and thus blocks it.

branching into known subfamilies: chromobox proteins (CBX, red), chromodomain Y-linked proteins (CDY, yellow), chromodomain-helicase DNA-binding proteins (blue) and chromo-barrel domain proteins (green). The GEN1 chromodomain was found to be distantly related to the CDY chromodomains and chromobox proteins, particularly to the chromo-shadow domains of CBX1, CBX3 and CBX5. This agrees with the result from the DALI search, in which CBX chromo-shadow domains and homologs thereof were among the closest structural matches. Together with the observed differences in residues forming the aromatic cage, it indicates that the GEN1 chromodomain forms a new subgroup with features from chromo- and chromo-shadow domains that emerged from a common ancestor within CBX/CDY proteins.

## GEN1-DNA interactions

The GEN1-HJ structure revealed that the upstream DNA-binding interface acts as a docking site for double-stranded DNA and that the chromodomain secures its position. The DNA is bound at the upstream interface and the hydrophobic wedge but does not extend into the active site or to the downstream interface (*Figure 1B/C/D*). Comparison of the structure of GEN1 to related structures of FEN1, Rad2 and EXO1 (*Miętus et al., 2014*; *Orans et al., 2011*; *Tsutakawa et al., 2011*) suggests that a DNA substrate has to extend to the downstream interface to position a DNA strand for cleavage by the active site of GEN1 (*Figure 1B/C* and *Figure 1F*). In the GEN1 structure, the end of the DNA arm attaches to the hydrophobic wedge provided by parts of helices α2-α3 and their connecting loop (*Figure 4A/B*), forming van-der-Waals contacts with the first base pair, which docks perfectly onto the protruding curb of residues 41–51 (*Figure 4B*). The uncleaved DNA strand is further stabilized and its geometrical arrangement is fixed by the upstream DNA-binding interface. Particularly, the DNA is contacted by a β-pin (strands β6-β7; *Figure 4A/C*) from one side and by R54 and F58 (*Figure 4A/D*) from helix α3 together with Y370 and K374 (helix α15) from the opposite side (*Figure 4A/C*). The key residues in the β-pin are T171 that forms a hydrogen bridge to the phosphate of the first base (*Figure 4A*, 'G1') and M172 that makes a van-der-Waals contact to the DNA backbone at the second base (*Figure 4A*, 'A2'). R54 reaches into the DNA minor groove and forms a hydrogen bond with the ribose ring oxygen at the third base of the uncleaved strand and F58 packs against the same ribose moiety (*Figure 4C/D*). Y370 and K374 in α15 form hydrogen bonds to the backbone of the third base of the uncleaved DNA strand (*Figure 4D*, 'G3').

An additional interaction point is provided by a β-hairpin from the chromodomain (strands β8-β9), one DNA turn upstream of the hydrophobic wedge (*Figure 4A/E/F*). This β-hairpin interacts with the complementary DNA strand by matching the protein backbone (residues 406–411) to the contour of the DNA backbone in a sequence unspecific manner (*Figure 4A/E*). The side chains of K404 and R406 project out, and they are in hydrogen bonding distance to the DNA (*Figure 4E*). Remarkably, R408 forms a polar interaction with Q65, which establishes a connection between the DNA contact point at the chromodomain and the nuclease core (*Figure 4E*). The interactions at the chromodomain extend the upstream DNA-binding interface to cover a full DNA turn, reinforcing the binding.

The downstream binding interface can be inferred from other Rad2/XPG structures (*Figure 1C/F*) as the nuclease core is well conserved in GEN1, FEN1, Rad2 and EXO1 (root mean square deviations of 0.9–1.1 Å for 161 Cα atoms, respectively). The residues corresponding to the tip of the thumb (residues 79–92), which are disordered in the GEN1 structure, likely form helix α4 upon DNA binding to the downstream interface as seen in human FEN1 and EXO1 (*Orans et al., 2011*; *Tsutakawa et al., 2011*). The missing residues in GEN1 have 35.7% identity and 78.6% similarity (BLOSUM62 matrix) to the corresponding residues in FEN1 (90–103), which form helix α4 in the FEN1-DNA complex (compare *Figure 2*). The same region is disordered in FEN1 when no DNA is bound (*Sakurai et al., 2005*). This indicates that also GEN1 undergoes such a disorder-to-order transition to form an arch with helices α4 and α6 upon substrate binding (*Patel et al., 2012*) and similar to the arrangement in T5 FEN (*Ceska et al., 1996*).

## The activity of GEN1 depends on correct DNA positioning

GEN1 has versatile substrate recognition features, ranging from gaps, flaps, replication fork intermediates to HJs (*Ip et al., 2008*; *Ishikawa et al., 2004*; *Kanai et al., 2007*). To understand the functional relevance of the GEN1 structure for DNA recognition we performed a series of mutagenesis studies with single point mutations and truncated protein variants (*Figure 5* and *Figure 5—figure supplement 1/2*) to investigate the effect on the active site (D30N), upstream DNA binding (R54E), downstream DNA binding (C36E), arch at the downstream interface (R89E, R93E, H109E, F110E), and chromodomain (Δchromo, K404E, R406E). We performed nuclease assays by titrating different amounts of GEN1 to a fixed DNA concentration of 40 nM for 15 min and DNA cleavage products were analyzed by native electrophoresis (*Figure 5A* and *Figure 5—figure supplement 1/2*). We used an immobile HJ and a 5' flap substrate side-by-side to facilitate the comparison of the effects on separate GEN1 functions. Notably, stoichiometric amounts of GEN1 were required to cleave HJ substrates whereas 5' flaps were readily processed with catalytic amounts (*Figure 5A*).

The active site modification D30N showed that the cleavage activity on both HJ and 5' flap substrates was lost in agreement with previously published data (*Ip et al., 2008*). According to our

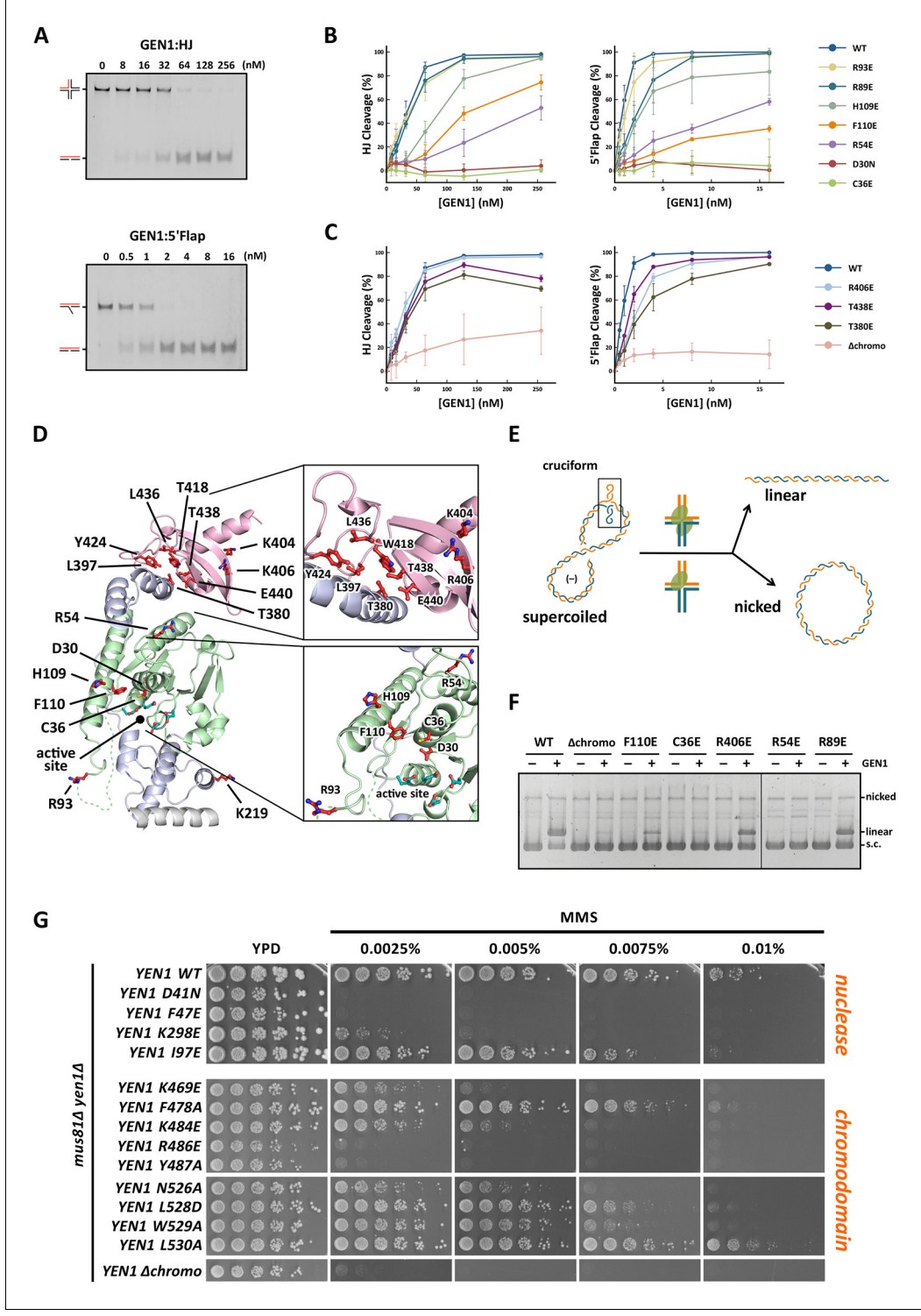

**Figure 5.** Functional analysis of GEN1. (**A**) Nuclease activity of GEN1 with HJ and 5'flap DNA. 40 nM 5' 6FAM-labeled substrates were mixed with indicated amounts of GEN1. Reactions were carried out at 37°C for 15 min, products were separated by native PAGE and analyzed with a phosphoimager. *Figure 5—source data 1* gives the sequences of DNA oligos used in biochemical assays and *Figure 5—source data 3* shows activity measurements. (**B**) Quantification of nuclease assays of wild type GEN1 and variants with mutated residues located at the protein-DNA interfaces. Percentage of cleavage was plotted against the enzyme concentration. Error bars depict the standard deviation calculated from at least three independent experiments. *Figure 5—figure supplement 1* shows representative gels from the PAGE analysis. (**C**) Quantification of nuclease assays of wild type GEN1 and

*Figure 5 continued on next page*

*Figure 5 continued*

variants with mutated residues located at the chromodomain. Error bars depict the standard deviation calculated from at least three independent experiments. *Figure 5—figure supplement 2* shows representative gels from the PAGE analysis. (**D**) GEN1 mutations used in this study. Locations of human GEN1 mutations used in biochemical assays and corresponding residues in yeast MMS survival assays are highlighted in red. Active site residues E134, E136, D155, D157 are marked in turquoise. (**E**) Schematic of the cruciform plasmid cleavage assay. A cruciform structure can be formed in plasmid pIRbke8$^{mut}$, which harbors an inverted-repeat sequence and is stabilized by negative supercoiling. Introducing two cuts across the junction point within the lifetime of the resolvase-junction complex yields linear products whereas sequential cleavage generates nicked products and the relaxed plasmid cannot be a substrate for the next cleavage. (**F**) Cruciform plasmid cleavage assay with different GEN1 variants. Plasmid pIRbke8$^{mut}$ was treated with 256 nM GEN1 each and reactions were carried out at 37°C for 15 min. Supercoiled, linear and nicked plasmids were separated by native agarose gel electrophoresis and visualized with SYBR safe under UV light. (**G**) MMS survival assays with yeast y*en1* variants. The survival of *yen1* mutants was tested under a *yen1Δ mus81Δ* background with indicated amounts of MMS. The top part shows mutations at GEN1-DNA interfaces and the bottom part mutations at the chromodomain (compare *Figure 5—figure supplement 3* for all controls and expression tests). *Figure 5—source data 2* gives a list of all yeast strains.

The following source data and figure supplements are available for figure 5:

**Source data 1.** Oligonucleotides used in biochemical assays.

**Source data 2.** Yeast strains used for MMS survival assays.

**Source data 3.** In vitro activity measurements of different GEN1$^{2-505}$ variants.

**Figure supplement 1.** DNA cleavage assays of different GEN1 mutations.

**Figure supplement 2.** DNA cleavage assays of different GEN1 fragments.

**Figure supplement 3.** MMS survival assays with yeast *yen1* mutants.

---

structure, R54 in helix α3 at the upstream interface fixes the substrate position by reaching into the minor DNA groove and we observed that R54E had a strongly reduced cleavage activity (~50%; *Figure 5B*), indicating a key role in substrate positioning.

Residue C36 in helix α2 points towards the downstream interface and likely contacts the DNA upon binding (compare *Figure 5D*). The corresponding FEN1 Y40, is a key residue stacking with the -1 base of the 5′ flap at the FEN1 active site (*Tsutakawa et al., 2011*). Therefore, we tested the cleavage ability of a GEN1$^{C36E}$ and found that the mutant protein had completely lost its enzymatic activity for both, HJ and 5′ flap cleavage, to the same degree as the active site modification D30N (*Figure 5B*). This effect is stronger than for FEN1$^{Y40A}$, which showed only a partial loss in activity (*Tsutakawa et al., 2011*). Our results suggest that C36 provides a polar interface for orienting and guiding the cleaved strand towards the active site and the lower gateway.

We further tested a glutamate modification of the superfamily-conserved R89 and R93 located in the disordered part continuing to helix α6, presumably forming an arch (see above). The arch was shown to facilitate cleavage by clamping flap substrates in FEN1 and the modification R100A showed a strong decrease in the cleavage activity (*Patel et al., 2012*). The GEN1 R89E mutation, corresponding to residue R100 in FEN1, showed that the activity of GEN1 with a HJ substrate was not altered. In the case of a 5′ flap substrate, cleavage was slightly reduced and it reached to the full level at enzyme concentrations higher than 10 nM. The effect of the R93E modification was even less pronounced compared to R89E. In contrast, the cleavage of both 5′ flap and HJ substrates depended strongly on F110 at helix α6 (thumb), which points towards the active site. An F110E modification showed a reduction in cleavage by 25% for HJ substrates, and the effect was even stronger for 5′ flap substrates, where the activity is reduced by 65%. The equivalent position in FEN1 is V133 showing a critical involvement in stabilizing 5′ flap DNA by orienting the -1 nucleotide for catalysis (*Tsutakawa et al., 2011*). We have also tested the effect of modifying H109, which neighbors the critical F110. Even though it points away from the active site, a glutamate at this

position reduced 5' flap cleavage to 83% and HJ cleavage recovered only at high substrate concentrations of 256 nM. Overall, the results suggest that F110 has a key position for DNA recognition and processing.

## Coordinated cleavage of HJs

Classical HJ resolvases introduce two symmetrical incisions across the junction point by coordinating the action of two active sites. The first nick is rate-limiting and the second one takes place near-simultaneously and within the lifetime of the resolvase-DNA complex. This mechanism has been well studied for bacterial and bacteriophage HJ resolvases (*Fogg and Lilley, 2000*; *Giraud-Panis and Lilley, 1997*; *Pottmeyer and Kemper, 1992*; *Shah et al., 1997*). Hence, it is thought that also GEN1 dimerizes upon binding to HJ substrates as indicated by coordinated cleavage and by an increase in hydrodynamic radius compared to protein alone (*Chan and West, 2015*; *Rass et al., 2010*). In order to further examine the effect of GEN1 modifications on HJ cleavage, we used a cruciform plasmid cleavage assay to evaluate GEN1's nicking function, as illustrated in *Figure 5E*. Here, the plasmid pIRbke8$^{mut}$ served as a substrate that contains an inverted-repeat sequence extruding a cruciform structure when supercoiled (*Chan and West, 2015*; *Lilley, 1985*; *Rass et al., 2010*). Coordinated dual incision of the cruciform (by a dimer) leads to linear duplex products with slow migration, whereas uncoordinated cleavage (by monomeric enzymes) results in nicked plasmids that migrate even slower (*Figure 5F*). Cruciform structures are reabsorbed when the superhelical stress is released upon single nicking and the DNA cannot serve as a substrate anymore.

We observed that wild type GEN1 resolved cruciform structures into linear products (*Figure 5F*) in agreement with previous reports (*Chan and West, 2015*; *Rass et al., 2010*). GEN1$^{C36E}$ (downstream interface) and GEN1$^{R54E}$ (upstream interface) showed only residual activity confirming their importance for HJ cleavage. The cruciform cleavage by F110E (thumb) was strongly reduced in line with our nuclease assays using small DNA substrates (*Figure 5B*). GEN1$^{R89E}$ (disordered part of the arch) did not show any appreciable effect, which suggests that this part of the arch is not directly involved in HJ recognition. Taken together, our results suggest that the positioning of HJ junction substrates both at the upper and the lower gateway is critical for productive cleavage. Furthermore, none of the tested modifications at the different DNA interaction interfaces was able to uncouple the coordinated HJ cleavage.

## The chromodomain of GEN1 facilitates efficient substrate cleavage

Agreeing with the structural significance for DNA binding, the truncation of the chromodomain (Δchromo, residues 2-389) showed a severe reduction (~3-fold) in HJ cleavage activity whereas all longer GEN1 fragments containing the chromodomain (2-464, 2-505 and 2-551) showed full activity (*Figure 5—figure supplement 2*). Interestingly, the effect of the chromodomain truncation is even more pronounced for 5' flap DNA cleavage than for HJs, showing a 7-fold reduction compared to wild type (*Figure 5C*). The activity of GEN1 in the plasmid-based cruciform cleavage assay was also severely hampered in the absence of the chromodomain (*Figure 5F*) showing only a weak band for linear products and no increase for nicked plasmid, emphasizing the importance of the chromodomain for GEN1 activity.

Further, to test the influence of the positively charged side chains K404 and R406 on DNA binding, we introduced charge-reversal mutations to glutamates and assessed their nuclease activities. Even though K404 and R406 are within hydrogen-bonding distance to the DNA, K404E, and R406E showed no appreciable influence on GEN1's nuclease activity. Only a slight reduction in cleavage of 5' flap substrates was observed for GEN1$^{R406E}$, whereas the processing of HJ substrates was not altered significantly (*Figure 5C*). This reinforces the conclusion from our structural observations that the chromodomain and the DNA interact through their backbones via van-der-Waals interactions.

## Influence of phosphorylation-mimicking chromodomain modifications

PhosphoSitePlus (*Hornbeck et al., 2014*) lists two phosphorylation sites at residues T380 and T438 in GEN1 that were found in a T-cell leukemia and a glioblastoma cell line. These residues are located in helix α15 and at the rim of the aromatic cage, respectively. Both phosphorylation sites are positioned to interrupt hydrophobic interactions between helix α15 and the chromodomain (*Figure 5D* and *Figure 4F*). Therefore, we tested if the phosphorylation-mimicking modifications T380E and

T438E had an effect on GEN1's activity. At low enzyme concentrations (<50 nM) HJ cleavage was similar to that of wild-type protein but at high concentrations the activity declined to less than 80% (*Figure 5C*). For a 5' flap substrate, the assay showed consistently lower activity than wild type, recovering to about 80% cleavage at the highest enzyme concentration (*Figure 5C*). These results suggest that phosphorylation of GEN1 chromodomain residues may regulate DNA recognition and cleavage.

## Physiological relevance of GEN1 interactions

To test the physiological relevance of the identified GEN1-DNA interactions, we investigated the survival of *Saccharomyces cerevisiae* mutant strains expressing variants of Yen1 (GEN1 homolog) after treatment with the DNA-damaging agent MMS (*Figure 5G* and *Figure 5—figure supplement 3*/*source data 2*). All Yen1 variants were expressed to a similar degree as endogenous Yen1, which was confirmed by Western Blot analysis (*Figure 5—figure supplement 3*). Because of the functional overlap of Mus81 and Yen1 in HR (*Blanco et al., 2010*) a double knockout (*yen1Δ mus81Δ*) was used and complemented with different variants of *Yen1*.

The control strain, complemented with wild type Yen1, survived MMS concentrations of up to 0.01%, consistent with the described hypersensitivity of *mus81Δ* mutants (*Blanco et al., 2010*; *Interthal and Heyer, 2000*). In stark contrast, cells containing either the active site mutant Yen1-D41N (corresponding to GEN1$^{D30N}$) or the downstream interface mutant Yen1-F47E (corresponding to GEN1$^{C36E}$) did not grow even at an MMS concentration as low as 0.0025% (*Figure 5G*). After the expression of the upstream interface mutant Yen1-I97E (corresponding to GEN1$^{R54E}$) cells showed a slight but significant growth defect at high MMS concentrations (see panels for 0.0075% and 0.01% MMS in *Figure 5G*). These results are therefore consistent with the in vitro cleavage results carried out with GEN1 mutants and showing a reduction in activity for R54E and no activity for C36E (see *Figure 5C*). As a last mutant in the nuclease core, we tested the K298E mutation which is located in helix α10 of the H2TH motif in the downstream DNA-binding interface, and for which we were unable to obtain the corresponding GEN1$^{K219E}$ modification for cleavage assays (compare *Figure 5D*). This mutant displayed a strong sensitivity towards MMS but lower than the one observed for the catalytic mutant, indicating that the mutant was partially functional in yeast (*Figure 5G*).

We next investigated the effect of mutations in the aromatic cage of Yen1's chromodomain (compare *Figure 3*) and found that their severity was strongly position dependent. Mutation of R486E and Y487A in Yen1, both of which are located near the base of the cage, corresponding to the W418 position in GEN1 (see *Figure 3C*), showed a strong effect on MMS sensitivity (see *Figure 5G*), similar to the one observed for the catalytic mutant, presumably due to a dysfunctional chromodomain. In contrast, mutations located further outside of the core (F478A and K484E) led to a less pronounced MMS sensitivity. The same was true for the K469E variant, which corresponds to position R406 at the chromodomain-DNA interface in GEN1 (see *Figure 3A* and *5F*), and for residues at the rim of the chromodomain (*yen1-N526A, yen1-L528D* and *yen1-W529A*), consistent with our in vitro observation for GEN1$^{T438E}$ (slightly reduced activity, *Figure 5C*). No effect on MMS sensitivity was detected for *yen1-L530A*, which corresponds to a conserved glutamate in chromodomains (E440 in GEN1). Lastly, we found that the deletion of the chromodomain (Yen1-Δ452–560) lead to a severe phenotype comparable to the active site mutant Yen1-D41N (*Figure 5G* and *Figure 5—source data 2*). The Yen1 variant lacking the chromodomain was expressed to levels similar to the full-length protein and we therefore conclude that the chromodomain is crucial for the function of Yen1. Taken together, the functional data of Yen1 mutants in vivo and GEN1 mutants in vitro point towards an essential and evolutionary conserved role of the chromodomain in GEN1/Yen1 proteins.

# Discussion

## Implications of the chromodomain

The structure of the human GEN1 catalytic core provides the missing structural information in the Rad2/XPG family. The GEN1 structure complements recent reports on the structures of Rad2, EXO1 and FEN1, (*Miętus et al., 2014*; *Orans et al., 2011*; *Tsutakawa et al., 2011*). Thereby, it gives insights how relatively conserved nuclease domains recognize diverse substrates in a structure-

selective manner and act in different DNA maintenance pathways. In comparison with other Rad2/XPG nucleases, GEN1 shows many modifications on common structural themes that give the ability to recognize a diverse set of substrates including replication fork intermediates and HJs. The upstream DNA interface of GEN1 lacks the 'acid block' found in FEN1, instead it has a prominent groove at the same position (compare *Figure 1*, 'upper gate') with a strategically positioned R54 nearby. Furthermore, the helical arch in GEN1 misses helix α5, which forms a cap structure in FEN1 and EXO1 that stabilizes 5' overhangs for cleavage. These features have implications for the recognition and cleavage of HJ substrates (see below). The most striking difference to other Rad2/XPG family members is that the GEN1 nuclease core is extended by a chromodomain, which provides an additional DNA anchoring point for the upstream DNA-binding interface. The evolutionarily conserved chromodomain is important for efficient substrate cleavage as we showed using truncation and mutation analyses. This finding opens new perspectives for the regulation of GEN1 and for its interactions with other proteins. Chromodomains serve as chromatin-targeting modules (reviewed in *Blus et al., 2011*; *Eissenberg, 2012*; *Yap and Zhou, 2011*), general protein interaction elements (*Smothers and Henikoff, 2000*) as well as dimerization sites (*Canzio et al., 2011*; *Cowieson et al., 2000*; *Li et al., 2011*). These possibilities are particularly interesting, as chromatin targeting of proteins via chromodomains has been implicated in the DNA damage response. The chromatin remodeler CHD4 is recruited in response to DNA damage to decondense chromatin (reviewed in *O'Shaughnessy and Hendrich, 2013*; *Stanley et al., 2013*). The chromodomains in CHD4 distinguish the histone modifications H3K9me3 and H3K9ac and determine the way how downstream DSB repair takes place (*Ayrapetov et al., 2014*; *Price and D'Andrea, 2013*). It is plausible that GEN1 uses its chromodomain not only as a structural module to securely bind DNA but also for targeting or regulatory purposes. Even though it was not possible to find any binding partner with a series of tested histone tail peptides, we cannot exclude that the chromodomain is used as an interaction motif or chromatin reader. It will therefore be interesting to extend our interaction analysis to a larger number of peptides and proteins. Interestingly, the modifications GEN1[L397E] and GEN1[Y424A] at the rim of the chromodomain did not alter DNA cleavage activity (*Figure 5—figure supplement 1*), however, mutations of residues at the rim of Yen1's chromodomain show a phenotype, suggesting an additional role like binding to an endogenous factor.

Another intriguing aspect of the chromodomain is that the conserved T438 at the rim of the aromatic cage and T380 at the closing helix α15 are both part of a casein kinase II consensus sequence for phosphorylation (Ser/Thr-X-X-Asp/Glu). *Ayoub et al., 2008* showed that the analogous threonine in the chromodomain of CBX1 is phosphorylated in response to DNA damage and phosphorylation disrupts the binding to H3K9me. We observed a reduction in DNA cleavage activity for the phosphorylation mimicking mutations T380E and T438E, which may suggest a regulatory role. They might function together and in combination with other modifications to provide a way of functional switching at the chromodomain. Furthermore, *Blanco et al., 2014* and *Eissler et al., 2014* recently identified several CDK phosphorylation sites in an insertion in the Yen1 chromodomain which affects HJ cleavage and together with phosphorylation of a nuclear localization signal (NLS) in the regulatory domain restricts Yen1's activity to anaphase. The insertion is not found in other chromodomains and it is extended in Yen1 compared to GEN1, which is lacking these phosphorylation sites (compare *Figure 3A/B*). Notably, the activity of Yen1 is negatively regulated by CDK-dependent phosphorylation (*Blanco et al., 2014*; *Chan and West, 2014*; *Eissler et al., 2014*; *Matos et al., 2011*), suggesting that the chromodomain is targeted by cell cycle kinases. It also provides a likely explanation for the different regulatory mechanisms found in GEN1 and Yen1 (*Blanco and Matos, 2015*; *Chan and West, 2014*; *Matos and West, 2014*). Exploration of the regulatory function of the GEN1 chromodomain will be an important topic to follow up, and this may lead to the understanding of the precise regulation mechanism of GEN1 as well as its substrate recognition under physiological conditions.

It is noteworthy that our analysis also revealed that the human transcription modulator AEBP2, which is associated with the polycomb repression complex 2 (PRC2), contains a chromo-barrel domain, which, to our knowledge, has not been reported so far.

## Recognition of DNA substrates

The GEN1-DNA structure showed a considerable similarity to the other members of the Rad2/XPG family, and this facilitated the generation of a combined model to understand substrate recognition

of GEN1 (*Figure 6*). This was done by superimposing the protein part of the FEN1-DNA complex (PDB 3q8k) onto our GEN1 structure and extending the DNA accordingly (*Figure 6A/B*). Remarkably, the superimposition of the proteins aligns the DNA from the FEN1 structure in the same register as the DNA in the GEN1 complex at the upstream interface (*Figure 6A and 6B* insert). Furthermore, the free 5' and 3' ends of the double flap DNA from the FEN1 structure point towards the lower and the upper gateway in GEN1, respectively (*Figure 6B*). We extended the GEN1 structure by homology modeling of the disordered residues 79-92 (helix α4) in GEN1 (*Figure 6B*). In addition to the similarity of this part to FEN1, the model readily showed the arrangement forming an arch structure. This would explain why GEN1 recognizes 5' flap substrates efficiently, analogous to FEN1, as the arch can clamp a single-stranded DNA overhang for productive cleavage. This also explains why the F110E modification in the arch at helix α6 hampered 5' flap cleavage severely. The

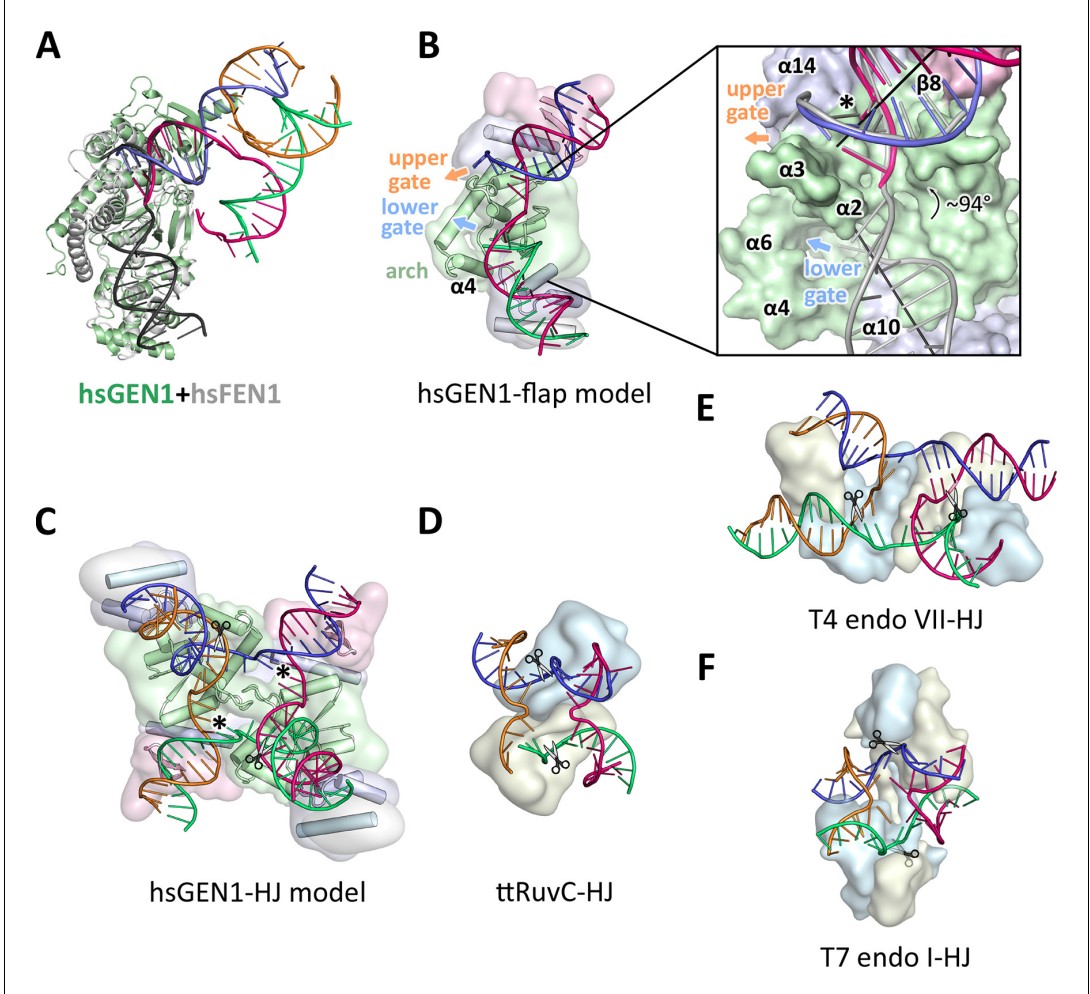

**Figure 6.** Substrate recognition features of GEN1. (**A**) Superposition of the protein part of the FEN1-DNA complex (PDB 3q8k, protein in gray, DNA in black) onto the GEN1-HJ complex (protein in green and the DNA strands in different colors). The FEN1-DNA aligns with the same register as the GEN1-DNA at the upstream interface. (**B**) Model for the recognition of a 5' flap substrate by GEN1. The DNA was extended using the superimposition from **A**. Homology modeling suggests an additional helix α4 (disordered residues 79–92) forming an arch with helix α6. The protein is shown in a simplified surface representation with the same colors as in *Figure 1* and structural elements are highlighted. The insert shows a zoomed in view of the hydrophobic wedge with the modeled FEN1-DNA in gray. (**C**) Model for the dimerization of GEN1 upon binding to a HJ substrate based on the 5' flap model in B. The monomers interlock via both arches (α4-α6) and the hydrophobic wedges (α2-α3) contact each other. (**D**) Structure of the *Thermus thermophilus* RuvC-HJ complex (PDB 4ld0). (**E**) Structure of the T4 endonuclease VII-HJ complex (PDB 2qnc). (**F**) Structure of the T7 endonuclease I-HJ complex (PDB 2pfj). Individual monomers are in surface representation, colored in light blue and beige, respectively. DNA strands are shown as ladders in different colors.

side chain points directly towards the active site and likely disturbs the stabilization of a 5' overhang for catalysis by charge repulsion. However, there are two features in GEN1 that vary from the arrangement in FEN1 and EXO1 considerably. Helix α6 is longer (24 instead of 15 residues) and helix α5 is missing in GEN1. As a result the arch points away from the DNA rather than forming a 'cap' structure as it is observed in FEN1 and EXO1 (*Orans et al., 2011*; *Tsutakawa et al., 2011*). Further-more, the modified arch in GEN1 provides an opening, marked as 'lower gate' in *Figure 6B*. These differences are likely the basis for GEN1's versatile DNA recognition features.

## Implications of an adjustable hatch in GEN1 for substrate discrimination

The diverging orientation of the arch (helices α4 and α6) in GEN1 compared to the one in FEN1 and EXO1 (helices α4, α5, and α6) may have thus significance for the recognition of HJ substrates. By pointing away from the active site the arch provides an opening to accommodate unpaired, single-stranded DNA to pass along the arch at the lower gate (groove between α2 and α4) (*Figure 6B* 'lower gate') from one GEN1 monomer to the upper gate (groove between α2-α3 and α14) (*Figure 6B* 'upper gate') of the other within a GEN1 dimer (*Figure 6B/C*). R54 is perfectly positioned at the minor groove to guide the second cleavage strand to pass through the upper gate (compare *Figure 4* and *Figure 6B/C*, marked with a asterisk). In FEN1, this position is occupied by the 'acid block', which stabilizes a single 3' flap of the unpaired substrate (*Tsutakawa et al., 2011*) and it would not accommodate longer 3' DNA overhangs. In our model, two GEN1 monomers come together crosswise upon HJ binding (*Figure 6C*). The helical arches of both proteins likely provide additional protein-protein interactions as well as protein-DNA contacts by packing against the back-bone of opposite DNA arms (*Figure 6C*). As a result, the GEN1 dimer orients both active sites sym-metrically across the junction point resembling the situation in bacterial RuvC (*Figure 6D*; *Bennett and West, 1995a*; *Górecka et al., 2013*). This arrangement would ensure that both inci-sions are introduced within the lifetime of the GEN1-HJ complex as observed biochemically by us and others (*Chan and West, 2015*; *Rass et al., 2010*). The mechanism likely works in a coordinated nick-and-counter-nick fashion, as shown for bacterial or bacteriophage HJ resolvases (*Fogg and Lil-ley, 2000*; *Giraud-Panis and Lilley, 1997*; *Pottmeyer and Kemper, 1992*; *Shah et al., 1997*) and recently for GEN1 (*Chan and West, 2015*).

The distance between both gates is bridged by unpaired bases in our GEN1-HJ model. This view is supported by the observation that FEN1 unpairs two bases near the active site through interac-tions with the hydrophobic wedge leading to strongly bent DNA arms between the upstream and downstream DNA interfaces. This mechanism seems to be a common feature of Rad2/XPG nucleases (*Finger et al., 2013*; *Grasby et al., 2012*; *Tsutakawa et al., 2011*). Consistent with this view, the bacterial RuvC resolvase (*Figure 6D*) has also been shown to unfold HJ junctions (*Bennett and West, 1995b*; *Górecka et al., 2013*). In the case of GEN1, the critical step would be the assembly of the dimer around the junction point in a highly restraint way and the introduction of the first nick. This releases the tension on the complex like a spring leading to an immediate second cut and sub-sequent disassembly of the GEN1-HJ complex. Furthermore, a HJ does not provide free DNA ends and adopts a structure that intrinsically restrains its degrees of freedom, thus inhibiting cleavage by a single GEN1 monomer. Altogether we speculate that the arch (helix α4-α6) acts like a lever or hatch switching between flap and HJ recognition modes. When a free 5' end is available it closes and clamps the flap, thus positions the DNA for cleavage. For the case of a HJ substrate, the arch adopts an open conformation, allowing unpaired, single-stranded DNA to pass, while preventing the correct positioning of the DNA for catalysis at first. HJ cleavage is inhibited until a second GEN1 monomer binds. This mechanism differs from the one used by bacterial or bacteriophage HJ resol-vases, which act as obligate dimers binding to DNA substrates in a concerted way (compare *Figure 6D–F*). Our model for DNA cleavage by GEN1 describes a conformational switch provided by a flexible arch that can discriminate between substrates containing free 5' ends or those with a restraint structure like HJs. This aspect may explain our observation that GEN1 cleaves 5' flap DNA catalytically while stoichiometric amounts are required for HJ substrates (*Figure 5A–C*). Using a switchable hatch in a spring-loaded mechanism would be an efficient way of preventing a single cut at a HJ junction while allowing GEN1 to adapt to recognize various DNA substrates and perform dif-ferent functional roles. Thus, GEN1 may have an intrinsic safety mechanism that ensures symmetrical dual incision across a branch point. Further studies have to address the exact engagement mechanism.

## GEN1 in a biological context

GEN1's biological role is not fully understood yet. Yeast cells are viable without the GEN1 homolog Yen1 even in the presence of DNA damaging agents as the Mus81-Eme1 complex can complement the defect (compare *Figure 5—figure supplement 3*; *Blanco et al., 2010*). Consistently, both proteins can cleave 5' flaps and HJ substrates in vitro. However, GEN1 can cleave intact HJs symmetrically whereas MUS81-EME1 is much more efficient with nicked DNA four-way junctions (*Castor et al., 2013*; *Wyatt et al., 2013*). *Matos et al., 2011* suggested that Yen1/GEN1 might serve as a backup enzyme to resolve persistent HJs that have eluded other mechanisms of joint molecule removal before cytokinesis.

Our analysis infers that HJ cleavage is slower than 5' flap cleavage (*Figure 5B/C*), bringing interesting implications for a safety control of GEN1's activity. GEN1 may have to assemble in an accurate way before it can cleave a HJ. Likewise, it increases GEN1's persistence time on HJs and opens a window for branch migration for extending the length of recombined stretches of DNA. Moreover, GEN1 recognizes various DNA substrates, which may point towards a general role in processing substrates in different DNA maintenance pathways. GEN1 has been shown to cleave replication fork intermediates, and it is implicated in the resolution of replication-induced HJs (*Garner et al., 2013*; *Sarbajna et al., 2014*). Like MUS81-EME1, it might also be important for the processing of fragile sites to ensure proper chromosome segregation (*Ying et al., 2013*). These functions have to be tested systematically to understand GEN1's biological role. In this context, the regulation of GEN1 is an important factor and needs to be explored. Our study identified a chromodomain extending the GEN1 nuclease core that might have a role in regulating the enzyme. An open question is the function and architecture of the remaining 444 amino acids at the C-terminus of GEN1. They are thought to regulate the nuclease activity and control subcellular localization (*Blanco et al., 2014*; *Chan and West, 2014*; *García-Luis et al., 2014*). It is very likely that new interaction sites and post-translational modifications in this region will be discovered in future. The presented structure together with additional studies will help to unravel these questions and to obtain a comprehensive view of the functions of the Rad2/XPG nucleases.

# Materials and methods

## Experimental procedures

### Protein expression and purification

Wild type human GEN1 and truncations thereof (residues 2-551, 2-505, 2-464, 2-389) were amplified by PCR from IMAGE clone 40125755 (Mammalian Gene collection, natural variant S92T, S310N, UniProtID Q17RS7) and cloned into a self-made ligation-independent cloning vector with various C-terminal tags followed by His8. Truncated versions were designed based on limited proteolysis in combination with domain prediction and functional assays to determine the smallest yet active fragment. The N-terminal methionine was cleaved by cellular methionyl-aminopeptidase, which is an essential requirement in the Rad2/XPG family as the N-terminus (conserved residue G2) folds towards the active site. Mutations were introduced by site-directed mutagenesis using Phusion Polymerase (NEB, Frankfurt/Main, Germany). All recombinant proteins were expressed in the *E. coli* BL21(DE3) pRIL strain (MerckMillipore, Darmstadt, Germany). Cells were grown at 37°C until mid-log phase and induced overnight with 0.2 mM IPTG at 16°C. Cells were harvested by centrifugation and resuspended in lysis buffer containing 1x phosphate buffered saline (PBS) with additional 500 mM NaCl, 10% (v/v) glycerol, 2 mM DTT, 1 mM EDTA, 1 µM leupeptin, 1 µM pepstatin A, 0.1 mM AEBSF and 2 µM aprotinin and lyzed by sonication. Cell debris was removed by centrifugation (75 600 *g* for 45 min), the clarified lysate was applied onto Complete HisTag Nickel resin (Roche Diagnostics, Mannheim, Germany) and washed with buffer A consisting of 20 mM Tris-HCl pH 7.5, 500 mM NaCl, 10% (v/v) glycerol, 2 mM DTT and followed by a chaperone wash step with 20 mM Tris-HCl pH 7.5, 500 mM NaCl, 2 mM ATP, 5 mM MgCl$_2$, 10% (v/v) glycerol and 2 mM DTT. The protein was eluted with buffer A containing 300 mM imidazole. The tag was cleaved, followed by cation exchange chromatography using a HiTrap SP HP column (GE Healthcare, Freiburg, Germany) with a linear gradient from 150 mM to 450 mM NaCl. Peak fractions were pooled and further purified by size-exclusion chromatography on a HiLoad 16/60 Superdex 200 (GE Healthcare) equilibrated with 20 mM Tris-HCl

pH 7.5, 100 mM NaCl, 5%(v/v) glycerol, 0.1 mM EDTA and 2 mM TCEP. Peak fractions were pooled, concentrated, flash-frozen in liquid nitrogen and stored at -80°C.

## Crystallization and data collection

GEN1[2-505 D30N] and DNA (4w1010-1 GAATTCCGGATTAGGGATGC, 4w1010-2 GCATCCCTAAGC TCCATCGT, 4w1010-3 ACGATGGAGCCGCTAGGCTC, 4w1010-4 GAGCCTAGCGTCCGGAATTC) were mixed at a molar ratio of 2:1.1 at a final protein concentration of 14 mg/ml including 1 mM MgCl$_2$ and co-crystallized by sitting drop vapor diffusion. Drops were set up by mixing sample with mother liquor consisting of 100 mM MES-NaOH pH 6.5 and 200 mM NaCl at a 2:1 ratio at room temperature. Crystals grew within 2 days, and several iterations of streak seeding were needed for obtaining diffraction quality crystals. For data collection, crystals were stepwise soaked in 10%, 20%, and 30% (v/v) glycerol in 100 mM MES-NaOH pH 6.5, 200 mM NaCl and 5% PEG 8000 and flash-frozen in liquid nitrogen. Diffraction data were collected at beamline PXII of the Swiss Light Source (SLS, Villigen, Switzerland) at 100 K with a Pilatus 6M detector. In order to obtain phase information, crystals were soaked for 10–30 min in 1 mM [Ta$_6$Br$_{12}$]Br$_2$, flash-frozen and data were collected at the Ta L(III)-edge. In addition, seleno-methionine (SeMet)-substituted protein was expressed in M9 media supplemented with SeMet, purified, and crystallized according to the protocol above and data were collected at the Se K-edge.

## Structure determination and refinement

All data were processed with XDS (*Table 1*, *Kabsch, 2010*). HKL2MAP (*Pape and Schneider, 2004*) found 12 tantalum and 8 selenium positions, which were used in a combined MIRAS strategy (multiple isomorphous replacement with anomalous scattering) in autoSHARP (*Vonrhein, et al., 2007*) to determine the structure of the GEN1-HJ complex. The obtained solvent-flattened experimental map was used to build a model with PHENIX (*Adams et al., 2010*) combined with manual building. The structure was then further refined by iterative rounds of manual building in COOT (*Emsley and Cowtan, 2004*), refinement with PHENIX and assisted by the PDB_REDO server (*Joosten, et al., 2014*). The structure was visualized and analyzed in PYMOL (*Delano, 2002*). Electrostatic surface potentials were calculated with PDB2PQR (*Dolinsky et al., 2004*) and APBS (*Baker et al., 2001*).

## Nuclease assay

All DNA substrates (*Figure 5—source data 1*) were synthesized by Eurofins/MWG (Ebersberg, Germany), resuspended in annealing buffer (20 mM Tris-HCl pH 8.0, 50 mM NaCl, 0.1 mM EDTA), annealed by heating to 85°C for 5 min and slow-cooling to room temperature. Different amounts of GEN1 proteins (as indicated) were mixed with 40 nM 6FAM-labeled DNA substrates in 20 mM Tris-HCl pH 8.0, 50 ng/µl bovine serum albumin (BSA) and 1 mM DTT. Reactions were initiated by adding 5 mM MgCl$_2$, incubated at 37°C for 15 min and terminated by adding 15 mM EDTA, 0.3% SDS and further, DNA substrates were deproteinized using 1 mg/ml proteinase K at 37°C for 15 min. Products were separated by 8% 1x TBE native polyacrylamide gel electrophoresis, the fluorescence signal detected with a Typhoon FLA 7000 phosphoimager (GE Healthcare), quantified with IMAGEJ (GE Healthcare) and visualized by GNUPLOT (*Williams et al., 2015*).

## Cruciform plasmid cleavage assay

The cruciform plasmid pIRbke8[mut] was a gift from Stephen West's lab (*Rass et al., 2010*), and it was originally prepared by David Lilley's lab (*Lilley, 1985*). 50 ng/µl plasmid were mixed with 20 mM Tris-HCl pH 8.0, 50 mM potassium glutamate, 5 mM MgCl$_2$, 50 ng/µl BSA and 1 mM DTT and pre-warmed at 37°C for 1 hr to induce the formation of a cruciform structure. Reactions were initiated by adding indicated amounts of GEN1, incubated at 37°C for 15 min and stopped as for DNA cleavage assays. The products were separated by 1% 1xTBE native agarose gel electrophoresis, stained with SYBR safe (Life Technologies, Darmstadt, Germany) and visualized under UV light.

## Sequence alignments and phylogenetic analysis

Sequences of GEN1 proteins from different organisms as well as all human chromodomain proteins were aligned to the human GEN1 sequence using the programs HHPRED (*Söding et al., 2005*), PSI-BLAST and further by manual adjustments. Alignments were tested by back-searches against RefSeq

or HMM databases. A phylogenetic tree was calculated by the program PHYML with 100 bootstraps using the alignment in *Figure 3—figure supplement 1* and a BLOSUM62 substitution model. The tree was displayed with DENDROSCOPE (*Huson and Scornavacca, 2012*).

### Histone peptide pull-down assay

The GEN1 chromodomain with a C-terminal His8-tag was immobilized on complete HisTag Nickel resin and washed twice with binding buffer consisting of 20 mM Tris-HCl pH 7.5, 200 mM NaCl, 5% glycerol, 0.1 mM EDTA, 0.05% (v/v) Tween-20 and 2 mM TCEP. Peptide mixtures containing 0.4 μM fluorescein labeled histone peptides were incubated with beads at 4°C for 1 hr and washed twice with binding buffer. Immobilized proteins were eluted with binding buffer supplemented with 300 mM imidazole and separated on 20% SDS-PAGE. Fluorescein-labeled peptides were visualized by detecting the fluorescence signal with a Typhoon FLA 7000 phosphoimager (GE Healthcare).

### Yeast genetics and MMS survival assay in *Saccharomyces cerevisiae*

All yeast strains are based on W303 Rad5+ (see *Figure 5—source data 2* for a complete list). *yen1△* or *yen1△ mus81△* strains were transformed with an integrative plasmid expressing mutant versions of *YEN1*. Freshly grown over-night cultures were diluted to $1 \times 10^7$ cells/ml. 5-fold serial dilutions were spotted on YPD plates with/without MMS (methyl methanesulphonate, concentrations as indicated) and incubated for 2 days at 30°C. The expression of 3FLAG-tagged Yen1 constructs was verified by SDS-PAGE and Western Blot analysis. Proteins were detected using a mouse monoclonal anti-FLAG M2-peroxidase (HRP) antibody (Sigma-Aldrich, München, Germany).

### Database entry

The coordinates of the human GEN1-Holliday junction complex have been deposited in the Protein Data Bank (PDB code 5t9j).

## Acknowledgements

We would like to thank Naoko Mizuno, Michael J. Taschner and Esben Lorentzen for many scientific discussions and critical reading of the manuscript. We are grateful for the support by the Department of Structural Cell Biology at the MPI of Biochemistry (MPIB), particularly for the help with screening by the Crystallization Facility and Claire Basquin for assistance with biophysical analysis. The Microchemistry Core Facility of the MPIB provided mass spectrometry analysis. We would like to thank Jürg Müller for providing histone tail peptides for binding tests and Stephen C. West for providing the plasmid pIRbke8mut for functional assays. Furthermore, we would like to acknowledge the professional assistance of the beamline staff at the Swiss Light Source, Villigen, Switzerland, during data collection. The Max Planck Society for the Advancement of Science supported this research through the Otto Hahn Program (BP) and the Max Planck Research Group Leader Program (CB).

## Additional information

### Funding

| Funder | Grant reference number | Author |
|---|---|---|
| Max-Planck-Gesellschaft | Max Planck Research Group Leader Program | Shun-Hsiao Lee Maren Felizitas Klügel Christian Biertümpfel |
| Max-Planck-Gesellschaft | Otto Hahn Program | Lissa Nicola Princz Boris Pfander |
| Max-Planck-Gesellschaft | MPI of Biochemistry | Bianca Habermann |

The funders had no role in study design, data collection and interpretation, or the decision to submit the work for publication.

## Author contributions
S-HL, CB, Conception and design, Acquisition of data, Analysis and interpretation of data, Drafting or revising the article, Contributed unpublished essential data or reagents; LNP, MFK, Acquisition of data, Contributed unpublished essential data or reagents; BH, Acquisition of data, Analysis and interpretation of data; BP, Acquisition of data, Analysis and interpretation of data, Drafting or revising the article

## Additional files

### Major datasets

The following dataset was generated:

| Author(s) | Year | Dataset title | Dataset URL | Database, license, and accessibility information |
| --- | --- | --- | --- | --- |
| Lee S-H, Biertump-fel C | 2016 | Crystal Structure of human GEN1 in complex with Holliday junction DNA in the upper interface | http://www.rcsb.org/pdb/explore/explore.do?structureId=5T9J | Publicly available at the RCSB Protein Data Bank (accession no. 5T9J) |

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
