## [Decision Letter]

Thank you for submitting your work entitled "Human Holliday junction resolvase GEN1 uses a chromodomain for efficient DNA recognition and cleavage" for consideration by *eLife*. Your article has been reviewed by three peer reviewers, one of whom is a member of our Board of Reviewing Editors (Volker Dötsch). The following reviewer has agreed to reveal their identity: Stephen West. The evaluation has been overseen by the Reviewing Editor and Ivan Dikic as the Senior Editor.

The reviewers have discussed the reviews with one another and the Reviewing editor has drafted this decision to help you prepare a revised submission.

Summary:

This is a very interesting paper that describes the structure of the N-terminal active site region of human GEN1 (residues 2-505) in complex with a Holliday junction. As expected, the authors found that the core of GEN1 is similar to that of other XPG/Rad2 nucleases. However, GEN1 contains a chromodomain as an additional DNA interaction site. The chromodomain of GEN1 is required for its full catalytic activity in vitro and is important for the in vivo function of the yeast homolog of GEN1, Yen1.

All reviewers agree that this manuscript should be published in *eLife*. The reviewers do not insist on any additional experimental work but would like the following questions being addressed in a revised manuscript.

Essential revisions:

1) In Figure 3, it would be good to include the CtGEN1 from *C. thermophilum* (Freeman et al., 2014 JMB) to see whether this newly identified GEN1 homolog also contain the chromodomain.

2) Mutation in the helical arch (R89E) does not have any appreciable effect. Given the potential involvement of the arch in the function of GEN1 suggested by the authors, have the authors tested more mutations in that region?

3) The chromodomain of Yen1 contains an insert, which has unknown function. Within the insert, there are a number of CDK phosphorylation sites that have previously identified (Blanco et al, 2014 Mol Cell; Eissler et al., 2014 Mol Cell). CDK phosphorylation of Yen1 has been shown to reduce DNA binding and catalytic activity of Yen1. This should be discussed.

4) As mentioned in the Discussion, the authors suggest that both protein-protein interaction and protein-DNA interaction are likely to be involved in GEN1 dimerisation upon HJ binding. Did the authors identify the potential dimerisation domain?

5) Unfortunately the authors did not carry out any biochemical experiments with the Yen1 mutants. It is impossible to know whether the introduced mutations impair nuclease activity or something as basic as protein stability. Have the authors performed Western Blot analysis or other assays for further biochemical characterization?

6) Is the fragment of GEN1 used for the structural studies able to support GEN1's in vivo roles? Even if it does not, which would not be completely unexpected, the information would be very useful.

---

## [Author Response]

In response to the comments, the revised manuscript contains additional data as follows:

Figure 3: We have included CtGEN1 in the sequence alignment of GEN1 chromodomains.

Figure 5: We have included two additional GEN1 mutations in the arch region (R93E and H109E).

Figure 5: We have added Yen1(Δchromo) in the MMS survival assays.

[Supplementary-material SD4-data]: We have confirmed the expression of Yen1 in all yeast strains that were used in MMS survival assays.

*1) In Figure 3, it would be good to include the CtGEN1 from C. thermophilum (Freeman et al., 2014 JMB) to see whether this newly identified GEN1 homolog also contain the chromodomain.*

We have extended the sequence alignment as suggested (Figure 3). Indeed, CtGEN1 also contains a chromodomain, suggesting a widespread evolutionary role for the function of Yen1/GEN1.

*2) Mutation in the helical arch (R89E) does not have any appreciable effect. Given the potential involvement of the arch in the function of GEN1 suggested by the authors, have the authors tested more mutations in that region?*

We have added data for two additional mutations in the arch area (see Figure 5). R93E (at the end of putative helix α4) showed a similar activity as the wt enzyme and H109E (in helix α6, next to the important F110 and pointing away from active site) showed reduced activity. This underlines the importance of helix α6 for catalytic activity and suggests a minor role for helix α4.

*3) The chromodomain of Yen1 contains an insert, which has unknown function. Within the insert, there are a number of CDK phosphorylation sites that have previously identified (Blanco et al, 2014 Mol Cell; Eissler et al., 2014 Mol Cell). CDK phosphorylation of Yen1 has been shown to reduce DNA binding and catalytic activity of Yen1. This should be discussed.*

Thank you for this insightful suggestion, we have added this point to the Discussion. It is indeed interesting that the chromodomain of Yen1 is targeted by CDK phosphorylation in order to modulate Yen1 activity. This insertion is smaller and lacks these phosphorylation sites (compare Figure 3) in GEN1 proteins outside the fungal kingdom. It offers a potential explanation for the yeast-specific regulation of Yen1 activity.

*4) As mentioned in the Discussion, the authors suggest that both protein-protein interaction and protein-DNA interaction are likely to be involved in GEN1 dimerisation upon HJ binding. Did the authors identify the potential dimerisation domain?*

So far, we were not able to determine the dimerization interface conclusively. In our hands, human GEN1 does not behave well under high concentrations that are necessary for most methods to characterize oligomerization. We have started experimental work using a systematic mutational analysis as well as optimizing conditions for the biophysical characterization of human GEN1. The analysis will take more time and we are planning to address this point in a follow-up study. However, we have added a citation for a recent publication from the West lab (Chan et al., 2015) to the relevant section in the Results. The study confirms GEN1’s dimerization upon binding to HJs by characterizing hydrodynamic properties of human GEN1 and by analyzing the coordinated cleavage of HJs.

*5) Unfortunately the authors did not carry out any biochemical experiments with the Yen1 mutants. It is impossible to know whether the introduced mutations impair nuclease activity or something as basic as protein stability. Have the authors performed Western Blot analysis or other assays for further biochemical characterization?*

This is an important point and we also felt that it had to be addressed for a conclusive analysis. Therefore, we undertook the effort to incorporate a C-terminal 3FLAG-tag into all Yen1 variants used in the MMS survival assays, as no antibody against Yen1 was available. We show in [Supplementary-material SD4-data] by Western Blot analysis that all mutants but R517A were expressed to levels similar to wt. Hence, we have removed mutant R517A from the manuscript. It is located in an insertion region of the Yen1 chromodomain that does not seem to be directly important for the function of Yen1.

Overall, these control experiments suggest that protein stability is not affected with different Yen1 variants, but rather that they have a more specific defect in Yen1 function, either by directly influencing DNA binding, catalytic activity or by influencing its regulation.

Moreover, we have also introduced a Yen1 variant lacking the chromodomain (Δchromo), which despite being expressed similarly as the full-length protein fails to rescue the MMS hypersensitivity of the yen1Δ knock-out. We therefore conclude that the chromodomain is essential for Yen1’s function.

*6) Is the fragment of GEN1 used for the structural studies able to support GEN1's in vivo roles? Even if it does not, which would not be completely unexpected, the information would be very useful.*

We have used the yeast system to address this question and have expressed two Yen1 truncations, 2-451 (without chromodomain) and 2-559 (with chromodomain). The latter construct corresponds to the GEN1 fragment used for structural studies. The truncations showed a severe MMS hypersensitivity ([Supplementary-material SD4-data]). This suggests that the C-terminal regulatory domain of Yen1 is critical for its function, in line with the reviewer’s prediction. We are currently following up on this point.